# Lower Nitrogen Availability Enhances Resistance to Whiteflies in Tomato

**DOI:** 10.3390/plants9091096

**Published:** 2020-08-26

**Authors:** Sreedevi Ramachandran, Sylvie Renault, John Markham, Jaime Verdugo, Marta Albornoz, Germán Avila-Sakar

**Affiliations:** 1Department of Biology, The University of Winnipeg, Winnipeg, MB R3B 2G3, Canada; sreedevi.ren@gmail.com; 2Department of Plant Agriculture, University of Guelph, Guelph, ON N1G 2W1, Canada; 3Department of Biological Sciences, University of Manitoba, Winnipeg, MB R3T 2N2, Canada; Sylvie.Renault@umanitoba.ca (S.R.); john.markham@umanitoba.ca (J.M.); 4Escuela de Pedagogía en Ciencias Naturales y Exactas, Facultad de Ciencias de la Educación, Universidad de Talca, Linares 3580000, Chile; jverdugo@utalca.cl; 5Centro Regional de Investigación e Innovación para la Sostenibilidad de la Agricultura y los Territorios Rurales, Centro Ceres, Pontificia Universidad Católica de Valparaíso, Quillota 2260000, Chile; malbornoz@centroceres.cl

**Keywords:** choice assays, defense, fertilizer, herbivore preference, herbivory, resistance, *Solanum*, tolerance

## Abstract

Soil nitrogen (N) supplementation via fertilizers may increase crop yields substantially. However, by increasing tissue N content, added N can make plants more attractive to herbivores, effectively reducing their resistance to herbivores (ability to avoid herbivore damage). In turn, greater pest infestation may cause more severe reductions in fruit production than a moderate N scarcity. In this study, we tested whether lower N supplementation results in greater resistance to whiteflies and lower fruit production in four tomato varieties. We assessed the effects of N availability on tolerance to herbivores (degree to which fitness is affected by damage) and tested for the long-hypothesized trade-off between resistance and tolerance. Plants grown at half of an agronomically recommended amount of N had greater resistance without a significant drop in fruit production. Tomato varieties differed in resistance and tolerance to whiteflies, and showed a clear trade-off between these modes of defense. Root:shoot ratios were greater at lower N, but had no clear relation to tolerance. We estimated that the economic benefit of decreasing N addition almost fully compensates for losses due to lower tomato production. Additionally, lower fertilization rates would contribute to reduce environmental costs of large-scale use of agrochemicals.

## 1. Introduction

Plant defense against herbivores may be effected through both resistance and tolerance traits [1]. Resistance traits are those that reduce the amount of plant tissue removed by herbivores such as trichomes, thick and tough cuticles, lignified cell walls, thorns, secondary metabolites (some of which are produced and stored in trichomes) with toxic or anti-digestive properties, and volatiles that attract natural enemies of herbivores [1,2,3]. Tolerance traits are those that reduce the detrimental effects of herbivore damage on plant fitness, and are still less well-understood compared to resistance traits (but see [4]). Tolerance traits include increased photosynthetic efficiency, meristem availability, and the capacity to store and mobilize resources such as carbohydrates and nitrogenous compounds [5,6,7]. Plant tolerance of herbivory may vary depending on the extent of the damage, kind of tissue or organ damaged, plant ontogenetic stage, costs of tolerance traits, and the evolutionary history of the interactions between plants and the different herbivore species that feed on them [8,9,10]. For instance, in *Arabidopsis thaliana*, stem mass (mostly corresponding to the inflorescence) and allocation to root growth have been found to correlate positively with tolerance of herbivory [6,11].

Several factors may alter the effectiveness of natural plant defenses against herbivores. Despite the well-known beneficial effects of N addition on plant growth and fruit production [12,13,14,15], increased N availability may, in fact, decrease plant resistance against herbivores if herbivores prefer to feed on plant tissues with high N content [16,17,18,19]. Such a preference would be expected based on the importance of N for herbivore cellular structure and metabolism compounded with the scarcity of the element in plant tissues relative to those of herbivores [20] and the better performance of herbivores that feed on N-rich plant tissues [21]. However, herbivore preference for N-rich tissues may not be universal, and could be influenced and even counterposed by several factors [22]. Indeed, in a critical examination of recent studies that addressed the influence of plant nitrogen content on herbivore feeding preference on terrestrial systems we found very few that clearly provided experimental evidence of herbivore preference based on plant tissue N content (Appendix A), with a clear scarcity of studies on phloem suckers despite this guild’s importance in natural and managed ecosystems [23].

Nitrogen availability could also alter plant tolerance of herbivory. Nutrient availability has been proposed to increase tolerance because the construction of tissues following herbivore damage requires materials (elements) contained in nutrients [24]. While plants with greater access to stored carbon and nutrients may grow faster and compensate better for herbivory, it is unclear whether inherent (constitutive) faster growth rates provide plants with greater compensatory ability. In fact, one could expect tolerance to be higher in slower growing plants because tolerance is estimated in reference to undamaged individuals, whose gains in biomass per unit time (correlated to fitness) become smaller as the growth rate of an individual decreases [25,26]. Thus, if N addition results in greater plant growth rates, it could also decrease plant tolerance of herbivory. Interestingly, the same prediction arises from the Limiting Resource Model [27], according to which, the increase of N switches the limitation of plant fitness from N to carbon, and herbivory affects the uptake of C, thus resulting in greater negative effects of herbivory on fitness.

Lastly, N availability may influence tolerance and resistance indirectly through trade-offs between resistance and tolerance traits [28,29,30], or between either type of defense trait and growth or reproduction [31]. Alternatively, the two modes of defense can be complementary. For instance, resistance traits may indirectly lower disease incidence by maintaining disease vectors such as whiteflies below threshold infestation densities, while tolerance traits counteract the negative effects on plant growth and reproduction of non-vector pests that are able to overcome the resistance traits of a plant [32]. 

Thus, the main goal of this study was to address gaps in our knowledge of the effects of N on both tolerance and resistance of plants to whiteflies, phloem suckers that can be important pests and disease vectors in both crops and wild plant populations [33]. Tomato was selected as a study system because of its paramount importance as a crop [34] and because whiteflies can cause severe losses in production directly and indirectly, as vectors of viral diseases. In addition, we had access to genotypes (varieties) with a range of trichome densities and resistance to a generalist caterpillar [35,36,37], and for which there were preliminary data on resistance to whiteflies (J. Verdugo, pers. comm.). Therefore, we used the tomato–whitefly system to test the following hypotheses: (a) that N addition reduces resistance to whiteflies, (b) that given varietal differences in leaf trichome density and kind, resistance to whiteflies differs among tomato varieties, (c) that there is a trade-off between resistance and tolerance to whiteflies, and (d) given the aforementioned putative trade-off, that varieties also differ in their tolerance to whiteflies. Given the association between resource allocation to roots and tolerance [11] possibly related to the capacity of plants to sequester C and N in roots as a response to damage [38,39], we hypothesized that tomato plants with greater resource allocation to roots (i.e., greater root:shoot ratio) would be more tolerant to whiteflies. A second goal of our study was to assess the economic benefits (increased resistance to whiteflies) and costs (losses due to a purported lower fruit production) of reducing the rate of N addition on tomato crops.

## 2. Results

For simplicity, in this section we do not show statistics for the effects of two- and three-way interactions among the main factors in our models (whitefly presence, variety, N level) unless such effects were statistically significant. The full models are available as Appendix A, and are summarized in Table 1. 

Importantly, we did not detect statistically significant effects of whitefly presence on any of the plant performance, growth, or reproduction variables analyzed (Table 1). For brevity, we do not repeat this result for each of the variables analyzed.

### 2.1. Nitrogen and Chlorophyll Analysis 

Total leaf chlorophyll concentration was significantly lower for plants in the Low and Medium N treatments than for those grown at High N (*F*_2,36_ = 11.58, *p* < 0.001; Figure 1a). Among the four varieties, Afamia had the highest leaf chlorophyll concentration, while Seven had the lowest (*F*_3,36_ = 11.22, *p* < 0.001; Figure 1b). We found similar results separately for chlorophylls *a* and *b* (data not shown). These models did not include a whitefly effect, since whiteflies had not been introduced to cages yet. Accordingly with chlorophyll concentrations, leaf total N content differed significantly among N levels (Kruskal–Wallis χ^2^ = 6.911, d.f. = 2, *p* = 0.032) with medians of 8.3, 8.7, and 10.3 mg g^−1^ (mean ± s.e., hereafter: 8.6 ± 1.0, 8.8 ± 0.7, and 10.5 ± 0.6 mg g−1, n = 24 per treatment), respectively, for plants in the Low, Medium, and High N levels.

### 2.2. Biomass, Fruit and Seed Production

Total fruit fresh and dry weight (the total production of a plant) were lower for plants in Low N than for those in the other two N treatments (fresh weight: *F*_2,214_ = 20.17, *p* < 0.001; dry weight: *F*_2,214_ = 19.17, *p* < 0.001; Figure 2). Variety had no significant effects on either response variable (Table 1; Figure 2). 

In terms of individual fruit size (as measured by fresh weight), plants of two varieties (Conquistador and Patrón) produced 61–64% smaller fruits when grown under Low N, but this pattern was not seen in the other two varieties (N × variety interaction effect: *F*_6,203_ = 3.05, *p* = 0.007). Overall, fruits ranged in size from 7.2 to 133.6 g of fresh weight (47.5 ± 1.6; median = 41.3; *n* = 229).

Our analysis for seed production (number of seeds per plant) showed a marginally significant effect of both variety and N, with non-significant interaction effects (Table 1). A simpler model without the interaction term showed significant effects of N treatment (*F*_2,231_ = 3.21, *p* = 0.042), but not of variety (*F*_2,231_ = 2.40, *p* = 0.069; Table 1; Figure 3). Plants grown under Medium N produced significantly more seeds than those in Low N, while seed production of High N plants was intermediate between the other two N levels (Figure 3a). 

N treatment had no significant effect on total seed mass per plant (*F*_2,214_ = 1.43, *p* = 0.241; Figure 3c). In contrast, we detected a significant effect of variety, due mostly to Afamia producing lower total seed mass than the other varieties (*F*_3,214_ = 5.02, *p* = 0.002; Figure 3d). 

We found significant effects of variety (*F*_3,214_ = 2.94, *p* = 0.034) and N treatment (*F*_2,212_ = 68.03, *p* < 0.001) on total plant vegetative biomass (Table 1). Decreased N availability resulted in lower vegetative biomass (Figure 4a). Conquistador and Afamia plants were smaller than Patrón plants, and plants from variety Seven were intermediate in total vegetative biomass (Figure 4b). Variety and N treatment also had significant effects on the root:shoot ratio of plants (respectively, *F*_3,214_ = 27.35, *p* < 0.001; *F*_2,214_ = 14.22, *p* < 0.001): Low and Medium N plants had greater root:shoot ratios than High N plants (Figure 4c), due mostly to having greater root mass, and less to having a reduced shoot (data not shown). Plants of varieties Patrón and Seven had greater root:shoot ratios than Afamia and Conquistador (Figure 4d).

### 2.3. Seed Germination

Seed germination differed among varieties (Deviance = 16.29, d.f. = 3, *p* = 0.0010), but there were no effects of N (Deviance = 0.22, d.f. = 2, *p* = 0.895) or the variety-by-N treatment interaction (Deviance = 9.80, d.f. = 6, *p* = 0.133). More importantly, we did not find a relation between the number of seeds produced by a plant and the likelihood that those seeds germinated (seeds: Deviance = 1.48, d.f. = 1, *p* = 0.224), which means that seed production is an adequate estimator of plant fitness for the purpose of estimating compensatory ability. 

### 2.4. Resistance

We found significant effects of N treatment (*F*_2,209_ = 288.7, *p* < 0.0001), variety (*F*_3,209_ = 236.5 *p* < 0.0001), and the variety-by-N treatment interaction (*F*_6,209_ = 14.3, *p* < 0.0001) on tomato resistance to whiteflies. The blocking factor (the individual Petri plate) did not have a significant effect on resistance (*F*_19,209_ = 0.47, *p* = 0.972). While the magnitude of the N effect differed among varieties, plants in the Low N treatment consistently had the greatest resistance against whiteflies (Figure 5). Afamia was, overall, the most resistant variety: not a single whitefly was found on leaf disks obtained from plants of this variety in the Low N, and the mean resistance for plants in Medium N was not significantly different from unity (indicating maximum resistance). In all other varieties, mean resistance differed significantly among all three N levels. Conquistador was the least resistant variety, and also, the one in which plants were more than twice as resistant at Medium N (2.24-fold) than at High N. In the other three varieties, resistance under Medium N was 1.12–1.26 times greater than at High N (Figure 5). 

### 2.5. Tolerance

Compensatory ability differed significantly among the varieties studied (*F*_3,108_ = 5.526, *p* = 0.001; Figure 6), but we did not find significant effects of N treatment (*F*_2,108_ = 0.590, *p* = 0.556) or the variety-by-N treatment interaction (*F*_6,108_ = 0.893, *p* = 0.503). Notably, compensatory ability values were generally positive (overall, 1.624 ± 0.231, median = 1.066), with median values greater than zero for all varieties except Afamia (Wilcoxon signed rank tests: Afamia: V = 292.5, *p* = 0.1104, combined Conquistador, Patrón, and Seven: V = 3518, *p* < 0.0001). Using the mean values of resistance and compensatory ability for each variety, we found a highly significant negative correlation between resistance and compensatory ability (Pearson’s *r* = –0.9954, *p* = 0.0046, d.f. = 2; Figure 7). This correlation was marginally significant when tested using individual plant values of resistance and compensatory ability (Pearson’s *r* = −0.1694, *p* = 0.0642, d.f. = 118).

## 3. Discussion

### 3.1. Plant Resistance and Tolerance

Our study shows that decreasing N availability to half of the recommended application rate for tomato production resulted in greater resistance to whiteflies without a significant decrease in total fruit mass between the two highest levels of N availability used. We also found differences among varieties in resistance and compensatory ability that show a negative association between the two modes of defense. Varietal differences in resistance were consistent with preliminary findings (J. Verdugo Leal, unpub. data) whereby Afamia and Patrón plants were more resistant than those of Conquistador and Seven.

Nitrogen has been recognized as a growth- and yield-limiting factor for both crops and wild plants [40,41]. Thus, the addition of N to soils has been deemed necessary for increased crop yield. However, plants with greater N content in their tissues could be more attractive to herbivores [20], which led us to hypothesize that by decreasing N availability, resistance to whiteflies would increase. Indeed, our study showed greater resistance to whiteflies under reduced N availability, with the magnitude of the change in resistance greater in varieties with lower overall resistance. 

Two non-mutually exclusive reasons may explain the greater resistance of plants growing under reduced N availability seen in our study: first, given the relative scarcity of N in plant tissues and the large amount of N needed by herbivores to construct their inherently protein-rich tissues, natural selection should have favored the evolution of greater ability to detect plant tissues with higher N content and greater preference for them [20,42]. Second, plants growing under high N availability may allocate more of their N to growth (structural proteins, rubisco, and chlorophyll) and less to N-based defense metabolites [43,44,45,46]. Such a change in resource allocation should have greater effects on plants whose resistance depends largely on carbon-based (C-based) secondary metabolites that accumulate more when there is not enough N to grow fast [47]. While tomato contains both C-based (chlorogenic acid, rutin, kampferol-rutinoside) and N-based (tomatine) secondary metabolites, both kinds were shown to correlate positively with C:N ratio [48], indicating that under high C, more N is used for secondary metabolites, and conversely, under high N availability, N is primarily used in the construction of new tissues (growth). In our study, plants in the High N treatment indeed had the highest shoot biomass and also the highest chlorophyll and N content, and lowest resistance to whiteflies, as expected if N were used preferentially for growth. However, we did not measure secondary metabolite concentrations in our plants, and therefore, we cannot ascertain to what extent the accumulation of C-based defense metabolites under scarcity of N contributes towards the greater resistance of Low and Medium N plants compared to just the greater nutritious quality (N content) of the High N plant tissues. Whichever the mechanism through which the Medium and Low N plants attained greater resistance than High N plants, our results are consistent with two other studies that found that whiteflies prefer tomato plants with high N content [49,50]. 

In addition to resistance, we estimated compensatory ability of tomato to whitefly feeding as a measure of tolerance. Tolerance traits provide defense against herbivores to plants that have been damaged by herbivores via mechanisms that decrease the fitness consequences of damage. Tolerance mechanisms include increases in photosynthetic rate, branching, and resource translocation to and from below-ground tissues [5,11,51]. In the present study, we found significant differences among tomato varieties in their ability to compensate for whitefly attack. We expected plants with larger allocation to roots (higher root:shoot ratios) to have greater tolerance to whiteflies. However, we did not see such a correlation: the two varieties with the highest and lowest compensatory abilities had roughly the same root:shoot ratio, which was lower than that of the other two varieties. Thus, other traits, including physiological mechanisms involved in resource translocation must be involved in tomato tolerance to whiteflies. 

Notably, three varieties overcompensated and the other one fully compensated regardless of N level, indicating high tolerance to whiteflies at the experimental infestation level used. Overcompensation should have been reflected in a positive whitefly effect on seed production (at least on the three overcompensating varieties). However, the minimal number of degrees of freedom achieved with the split-plot design might have precluded the detection of such an effect on seed production and other response variables.

Interestingly, as per our third hypothesis, varietal differences in resistance and tolerance were negatively correlated, indicating a trade-off between resistance and tolerance to whiteflies among the four tomato varieties studied (Figure 7). A previous study on *Arabidopsis thaliana* in our lab found a similar trade-off in plants at the vegetative stage, that became weaker with ontogeny and was undetectable at the reproductive stage [9]. The expectation of such a trade-off between resistance and tolerance is based on a proposed functional redundancy of resistance and tolerance traits: they both confer defense against herbivores. However, resistance traits may defend plants better from generalist herbivores than from specialists, while tolerance traits would enable plants to recover from damage with less regard to the degree of dietary specialization of the herbivore or the mode of damage (e.g., loss of tissue to chewing herbivores or phloem contents to whiteflies), and even be useful in the recovery from damage due to abiotic agents. In summary, resistance and tolerance traits can be complementary [52,53]. 

### 3.2. Economic Analysis of Lowering Rates of Fertilization

Our findings have important implications for the design of policies and good practices of fertilizer use. In our study, plants grown with half of the recommended N level were, depending on their variety, 12–124% more resistant to whiteflies than those grown using the commercially recommended N level without a statistically significant decrease in total fruit mass produced. Only plants in the lowest N treatment level had a significantly lower total fruit fresh weight than the other two treatments. Similarly, other studies on tomato found yield not to differ between fertilization rates of 90 and 180 kg N ha^−1^ [54], and even not to increase with fertilization rates above 750 mg N per plant per week [55]. Moreover, greater accumulation and movement of residual soil N occurred when N was added at a rate of 180 kg N ha^−1^ compared to 90 kg N ha^−1^, which indicates that plants are not able to take up N at rates above 90 kg N ha^−1^ [54]. 

For the present study, it could be argued that the failure to detect a decrease in total fruit mass at the Medium N level is a problem of statistical power, and therefore, in practice, farmers might incur economic losses if, by applying half of the recommended amount of N, their yields are reduced. Therefore, it is important to weight the potential economic loss due to reduced N application (a 15% yield reduction as per our results for fruit production at Medium N) against the potential economic and environmental benefits. According to data from Statistics Canada, the 2011–2014 average yield of tomato was 68,880 kg ha^−1^, which brought an average of $11,194 CAD ha^−1^ to farmers (approx. $8631 USD, using the average of the annual mean exchange rates for 2017 and 2018: $1 USD = $1.297 CAD; www.bankofcanada.ca). A 15% loss would equate $1294 USD ha^−1^.

The loss calculated above already takes into account the main direct advantage of reduced fertilization: increased resistance. Greater resistance to whiteflies results in plants losing a smaller portion of their materials and energy directly to whiteflies, which could be reflected in greater yields. At this point we are uncertain of the extent to which the difference in yield between full and half the rate of N application may change with whitefly infestation intensity, but it is likely that the difference has a maximum at intermediate infestation levels because with less infestation, there is little advantage of resistance, and with too high an infestation, flies may resort to low-nutritional quality plants because of competition with whiteflies on high N plants, thus negating the potential resistance of plants with lower N content. However, an additional advantage brought about by enhanced resistance to whiteflies is the reduction in spread of viral diseases via whiteflies. This in turn, would favor greater yields, but estimating those additional economic benefits is out of the scope of the present study, and would require data on the epidemiology of viral diseases in tomato.

An immediate (gross) economic benefit of using half of the commercially recommended fertilizer application rate is a lower expenditure on N-based fertilizer. While the actual amount of N recommended may depend on the type of soil and other factors, we estimated these savings could amount to $276 ha^−1^ in Chile or $78 ha^−1^ in the U.S. (all figures in USD; Table 2). Meanwhile, the increased resistance against whiteflies attained by lowering N application should decrease the need for pesticides, and reduce the need to clean off the sooty mold that commonly develops on fruits of whitefly-infested plants [56]. We estimated the savings on pesticide application to be $692 ha^−1^, and those on fruit cleaning, $196 ha^−1^. Adding these savings to those on N, farmers could save $1164 ha^−1^, which compared to the $1184 ha^−1^ estimated loss in fruit production yields a net loss of only $20 ha^−1^ in Chile (the net amount depends on the actual yield per ha). While the actual net gains or losses may vary among regions and countries due to market fluctuations and differences in salaries, prices of agrochemicals, soil, and length of growing season, among other factors, the above estimates show that the economic benefits of decreasing N addition to half of what is currently recommended may almost fully compensate any losses in fruit production. 

In addition, reduced fertilizer and pesticide application rates may entail economic benefits in terms of recovered ecosystem services. Although sufficient organically available soil N is needed for adequate plant growth and fruit production, crop plants typically uptake only about 50% of the applied manure or fertilizer [57,58,59]. For instance, N uptake rates in tomato ranged from 32% to 53% in one study [60], and from 13% to 30% in another one [54]. Nitrogen that is not absorbed by plants ends up leaching from soil into groundwater or running off from agricultural fields by surface currents into water reservoirs, processes with important ecological consequences, including eutrophication and direct toxicity to aquatic biota [61,62]. Not surprisingly, N leaching from agricultural fields to groundwater is directly related to N fertilization rate [63]. The economic and environmental costs of current fertilization practices could be mitigated by increasing N uptake efficiency and reducing fertilizer application rates [64,65,66]. The economic value of ecosystem services that could be recovered by lowering fertilization rates could be estimated through valuations based on surveys (see [67]), and would contribute to offset any possible losses and serve as incentives for farmers and policy makers to reduce the recommended rates of fertilization. 

## 4. Materials and Methods

### 4.1. Study System

Tomato, *Solanum lycopersicum* L. (Solanaceae), is one of the most important commercial vegetable species in the world, and whiteflies (Hemiptera: Aleyrodidae, ca. 1500 species) are one of the leading causes of tomato yield loss worldwide, whether in greenhouses or agricultural field operations [33,68,69]. As phloem feeders, whiteflies cause general weakening and reduced growth of their host plants [70]. In addition, whiteflies are vectors of several kinds of viruses that cause diseases detrimental to plant growth, reproduction, and survival [71,72,73]. Control of whitefly infestations through pesticide application has had limited success [74,75] for several reasons, including the high population growth rates and rapid evolution of resistance to insecticides in whitefly populations, as well as the preference of whiteflies for the abaxial surface of leaves, where they are relatively protected from natural enemies and insecticides applied as aerosols [70,76,77]. 

### 4.2. Experimental Design 

Four commercial tomato varieties (Afamia, Conquistador, Patrón, and 7742-hereafter, Seven) were chosen based on preliminary evidence of variability in their levels of resistance to whiteflies (J. Verdugo Leal, unpub. data) under contrasting levels of drought stress, with Afamia and Patrón being more resistant than the other two varieties. 

A total of 240 plants (60 per variety, 20 per variety–N level–whitefly combination) were grown under one of three levels of soil N fertilization and two levels of whitefly infestation, and assessed for the effects of N supplementation and whitefly infestation on yield and resistance and tolerance to whiteflies after 22 weeks of growth. Plants were grown in a 4:1 mix of peat-based media (Pro-Mix BX, Premier Tech Ltd., Rivière-du-Loup, QC, Canada) and sand in 3.8 L pots in a greenhouse at 22.5 ± 1.9 °C and 64.1% ± 6.8% RH, from June to September 2016 and 20.3 ± 0.7 °C and 56.3% ± 3.6% RH from October to December. Whiteflies (*Bemisia tabaci*) were collected from tobacco plants at a different greenhouse and reared on tobacco plants inside anti-aphid screen cages in a growth chamber at 28 °C and 65% RH on a 16:8 h light:dark cycle.

One third of the plants of each variety (20) were randomly assigned to either a “High”, “Medium”, or “Low” N addition treatment. Initially, High N plants were scheduled to receive 112 kg N ha^−1^ during the growth season, which is an amount of N recommended for commercial tomato field production [13,78]; Medium N plants were to receive half of the recommended N, and Low N, would only receive N when more than half of the leaves on a plant showed chlorosis (yellowing) of ca. 25% of its leaf area. To supply these levels of N to the plants, we modified Hoagland’s nutrient solution [79] as follows: for High N, 4.45 mM Ca(NO_3_) _2_ (the sole source of N) was used; for Medium N, half the amount of Ca(NO_3_) _2_ was replaced with CaCl_2_; and for Low N, all Ca(NO_3_) _2_ was replaced with CaCl_2_ (Table 3). Fertilizer was added in four initial splits so as to meet the N requirement of tomato plants at different stages of growth. For High and Medium N plants, the first two splits were given on the 4th and 5th weeks of growth. The other two splits were applied when the plants started flowering and setting fruit (8th and 12th week of growth), the stage at which leaf N content drops rapidly [80,81]. In two occasions, Low N plants started to show chlorosis, so we fertilized them with the solution used for Medium N plants (applications on the 8th and 12th weeks of growth). By the end of the season, plants in the High and Medium N treatments started to show signs of chlorosis, so we gave them one extra split of 14 kg N ha^−1^ on the 17th week of growth. Thus, the total amount of N added to plants in the High, Medium, and Low N treatments was, respectively, 126, 70, and 28 kg N ha^−1^, with Medium and Low N plants receiving, respectively, 55.6% and 22.2% of the amount of N given to High N plants.

Plants were assigned random positions on one of two tables, with 10 plants per variety–N treatment combination on each table (120 plants per table). On the 9th week of growth, half of the plants in each variety–N treatment combination were randomly assigned to a whitefly infestation treatment, and the other half were left to grow without whiteflies (five plants per variety–N level–whitefly combination per table: five plants × four varieties × three N levels × two whitefly treatment = 120 plants). By then, plants in the Medium and High N treatments had received three applications of fertilizer, and those in Low N had received one. Due to the high cost and logistic difficulties of building separate cages for each plant inside which to include (or not) whiteflies, the whitefly infestation treatments (with/without) were applied to four groups of 60 plants as follows: each group was placed either inside a cage into which 1300 whiteflies were later introduced, or in a mock cage (half-open on two sides) without flies. In this way, each cage had five plants per variety–N level combination. Cages and mock cages were made of anti-aphid screen. Mock cages were used to provide all plants with similar light and humidity conditions regardless of the presence or absence of whiteflies. 

### 4.3. Nitrogen and Chlorophyll Analyses 

To confirm that leaf N content varied according to the N addition treatments, we sent samples of leaves collected after the second fertilizer application (5th week of the experiment, before whiteflies were introduced into the cages) for N quantification (^13^C and ^15^N isotope analyses of solid materials, Stable Isotope Facility, UC Davis, California). To keep the amount of leaf tissue removed to a minimum, only one leaflet from the most apical fully expanded leaf (length ≥ 90% of the longest leaf on the stem) was collected from each plant on the day its first flower opened. Leaflets were freeze-dried (Labconco, Missouri) for 24 h and stored at −20 °C until processed for analysis. A total of 72 samples (six individuals per variety–N level combination) were sent for N determination. 

We also determined leaf chlorophyll concentration [80,81] on the same leaves used for N determination to check whether it varied consistently with N availability [82,83]. Given the small amount of leaf biomass obtained from each plant, leaflets from five plants in each variety-by-N level combination were randomly selected and pooled so as to attain 1 mg samples of freeze-dried leaf material. Leaf samples were incubated in 6 mL methanol for 24 h in darkness. We collected the leaf extract in each vial and measured its absorbance at 650 and 665 nm using a spectrophotometer (Ultraspec, Massachusetts). We determined chlorophyll concentrations (mg chlorophyll g^−1^ leaf tissue by means of the McKinney equations [84,85]:
*Chlorophyll_a_* = 16.5*A*_665_ − 8.3*A*_650_
*Chlorophyll_b_* = −12.5*A*_665_ + 33.8*A*_650_
*Chlorophyll_a+b_* = 4*A*_665_ + 25.5*A*_650_
where *A*_665_ is the absorbance at 665 nm, *A*_650_ is the absorbance at 650 nm. 

### 4.4. Biomass, Fruit and Seed Production 

We collected fruits as they ripened and weighed them to obtain their fresh weight. Then we extracted their seeds, and dried both fruits and seeds to constant weight by placing them in a drying cabinet at 68 °C for 4 day. Fruit fresh weight relates more to measures of crop production, while dry weight, as a measure of fruit biomass, is more useful to describe resource allocation. Once dry, seeds were counted and weighed. 

After 22 weeks of growth, plant shoots were harvested by severing the stems at the base, 2.5 cm above the media. Shoots were split into stems and leaves. Roots were harvested after carefully washing off the growth media. All plant material was dried and weighed. 

### 4.5. Seed Germination 

Preliminary analyses revealed potentially important variation in seed size among individuals. Given the possibility of a trade-off between the number and size of seeds produced [86] that could alter the suitability of seed production as an estimate of individual plant fitness, we conducted a germination experiment using 10 seeds from a randomly selected fruit of every plant. Seeds were soaked in water for 12 h followed by a 5 min dip in 90% ethanol for external sterilization. Seeds were placed on plain agar media (8 g agar/L) in Petri plates in groups of four randomly selected plants per plate. Plates were sealed and kept in a dark growth chamber at 28 °C and 65% RH. Seed germination started 5 day after planting, and was monitored daily for another 15 day, after which, seeds that had not produced radicles were scored as not-germinated. 

### 4.6. Resistance 

To assess the effects of N treatment and tomato variety on plant constitutive resistance to whiteflies, we conducted a choice assay. Leaf disks were collected from the first fully expanded leaf from the apex of plants before they were exposed to whiteflies. Disks with an area of 1.77 cm^2^ were cut out using a cork borer. A total of 12 leaf disks, abaxial side up, (one each from plants of the 12 combinations of variety and N levels) were arranged randomly in a 10 cm diameter Petri plate. Fifteen whiteflies, reared on tobacco plants, were introduced to the Petri plate and allowed to choose between the disks. After 6, 12, and 24 h, we counted the number of whiteflies on each disk. Since the number of whiteflies on leaf disks did not change after 12 h, we used these numbers to estimate resistance as: $R_{ijk}=1-\frac{n_{ijk}}{n_{max}}$, where *R_ijk_* and *n_ijk_* are, respectively, the resistance score of, and the number of flies on disk *k* of variety *i* and N level *j*, and *n_max_* is the maximum number of whiteflies found on any of the disks examined. Therefore, this resistance score takes a value of zero if the number of flies found on a disk was equal to the maximum number of flies found on any disk, and one if no flies were found on the disk. This way of measuring resistance is consistent with Karban and Baldwin’s (1997) definition of resistance in terms of reduced herbivore preference: disks with many flies would get low resistance scores, indicating low plant resistance, while discs with few or no whiteflies would have high resistance scores, indicating high plant resistance. 

Twenty replicates of the assay (sets of 12 disks in a Petri plate) were conducted so as to include all the plants in the experiment. We analyzed the effects of N, variety, and their interaction on resistance by means of a GLM that included the particular Petri plate as a blocking factor. An alternative model using Poisson distribution and a correction for overdispersion showed consistent results. For clarity and brevity, we present only the GLM on the original resistance values. 

### 4.7. Tolerance

Tolerance can be estimated as the slope of the function between fitness and the amount or proportion of tissue damaged by herbivores. Being a norm of reaction, this estimate requires groups of closely related individuals. If we considered plants within a variety to be more closely related than among varieties, we could have four such groups, but this would result in low statistical power to detect slopes different from zero, and thus, to detect under-compensation (negative slope) or over-compensation (positive slope). To circumvent this problem, we estimated compensatory ability, a measurement at the individual level akin to tolerance, calculated as $CA_{ij}=\frac{S_{ij}}{{\tilde{S}}_{oj}}-1$, where *CA_ij_* and *S_ij_* are, respectively, the compensatory ability of and number of seeds produced by individual *i* in variety *j*, inside the full cage (i.e., with whiteflies) and ${\tilde{S}}_{oj}$ is the median number of seeds produced by plants of variety *j* grown in mock cages (i.e., without whiteflies) [6]. In this way, $CA_{ij}>0$ indicates over-compensation, and $CA_{ij}<0$ indicates under-compensation.

The effects of variety, N, and whitefly feeding on biomass of stem, leaves, and roots were used to elucidate the resource allocation patterns that favor tolerance to whiteflies, with the aim of improving our understanding of the mechanisms of tolerance. We found little variation among varieties or N levels in leaf or stem biomass or their sum (i.e., shoot biomass; data not shown), so we show here results for the root:shoot ratio as a summary of resource allocation patterns.

### 4.8. Statistical Analysis 

Since we had two cages with flies and two mock cages without flies, we used a split-plot design with N level and variety as fixed effects factors, and whitefly presence (sub-plot) nested within cage (plot) as a random factor. To estimate the effects of variety, N treatment, and whiteflies on total fresh and dry fruit mass, seed production, seed mass, leaf, stem and root biomass, root:shoot ratio, resistance, and compensatory ability, we used general linear models that included the main effects (variety, N treatment, and whiteflies) and their interactions. We used square root- and arcsine-transformed data, respectively, for seed production and compensatory ability to achieve normality and homoscedasticity of residuals. Similarly, we used a logarithmic transformation for fruit size (total fresh weight/number of fruits produced). For statistically significant effects, we assessed differences among means using Tukey’s multiple comparisons tests. 

All general linear models were analyzed using Minitab 16 (Minitab Inc., Pennsylvania). We used a generalized linear model with a logit link function to analyze seed germination as a binomial dependent variable (number of seeds germinated vs. not-germinated; R Core Team, 2018). 

## 5. Conclusions

In our study, reducing N availability increased tomato resistance to whiteflies, but it did not affect tolerance. When the soil N level was reduced to half of the commercially recommended amount for tomato, resistance to whiteflies increased between 12% and 124%, depending on tomato variety. Importantly, lower N availability only decreased fruit production significantly at the lowest level used (Low N treatment). Reducing the rate of N application to crops should result in lower costs of production, and could incentivize a decrease in the use of fertilizers and pesticides. 

## Figures and Tables

**Figure 1 plants-09-01096-f001:**
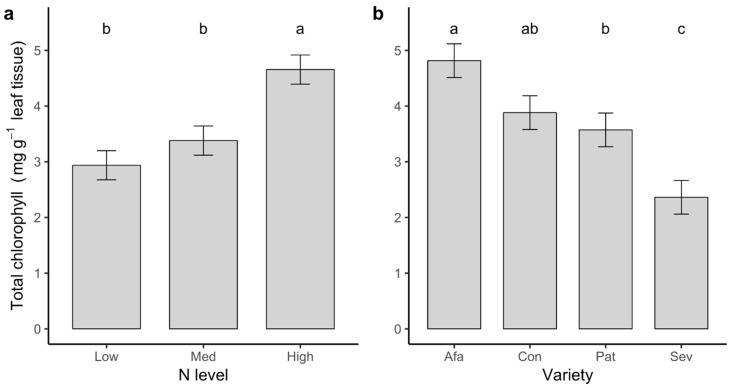
Least squares means ± 1 s.e. of total chlorophyll content per gram of freeze-dried leaf tissue collected from four tomato varieties grown at three N levels. (**a**) Means for plants grown at Low, Medium (Med), and High N level treatments. (**b**) Means for plants of the four different varieties (variety name abbreviations in this and other figures: Afa = Afamia, Con = Conquistador, Pat = Patrón, Sev = Seven). Fully expanded leaves for this analysis were collected at the time plants started flowering. Letters indicate significant differences among means according to Tukey’s multiple comparisons tests.

**Figure 2 plants-09-01096-f002:**
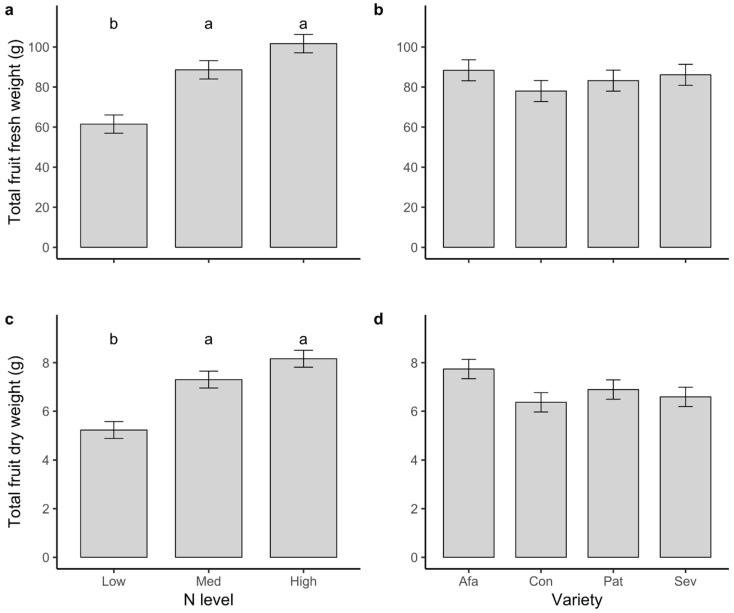
Least squares means ± 1 s.e. of (**a**,**b**) total fruit fresh weight and (**c**,**d**) total fruit dry weight per plant for plants of four tomato varieties grown at three N levels. Letters indicate significant differences among means according to a Tukey’s multiple comparisons test.

**Figure 3 plants-09-01096-f003:**
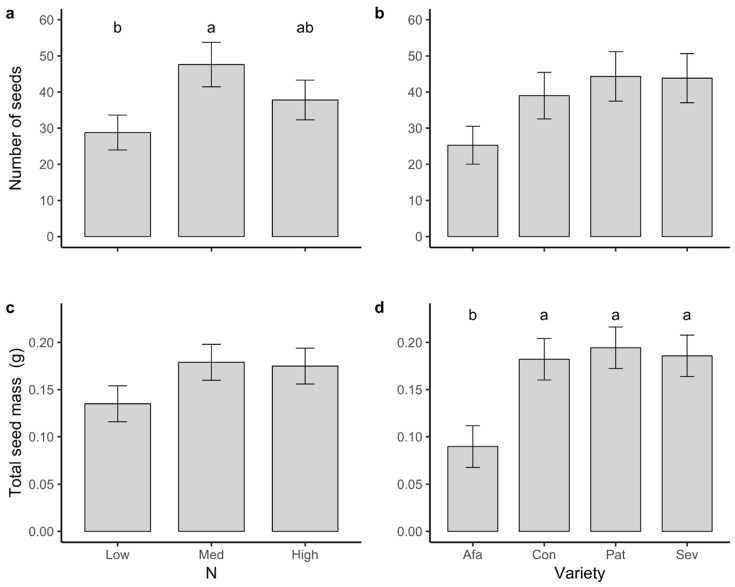
Least squares means ± 1 s.e. of (**a**,**b**) the number of seeds produced per plant and (**c**,**d**) total seed mass per plant of plants of four tomato varieties grown at three N levels. Letters indicate significant differences among means according to Tukey’s multiple comparisons tests.

**Figure 4 plants-09-01096-f004:**
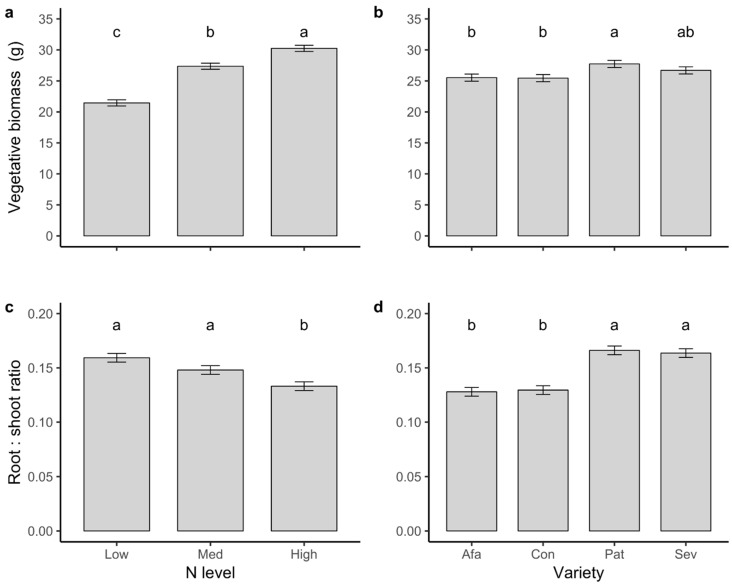
Least squares means ± 1 s.e. of (**a**,**b**) total vegetative biomass per plant (roots, shoots, and leaves) and (**c**,**d**) root:shoot ratio of plants of four tomato varieties grown at three N levels. Letters indicate significant differences among means according to Tukey’s multiple comparisons tests.

**Figure 5 plants-09-01096-f005:**
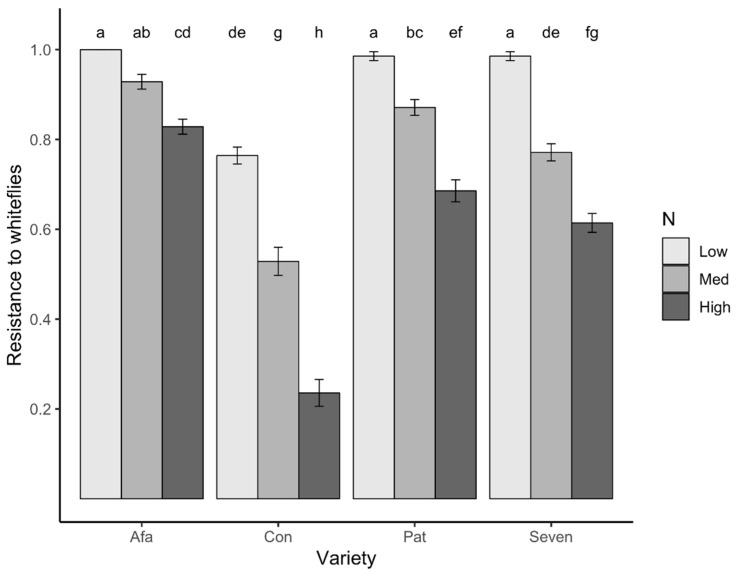
Least squares means ± 1 s.e. of resistance to whiteflies (back-transformed) of tomato plants from four varieties grown at three N levels. Resistance values were obtained from a whitefly choice assay using the number of whiteflies on leaf discs after 24 h (larger resistance values indicate greater resistance to whiteflies; see text for details).

**Figure 6 plants-09-01096-f006:**
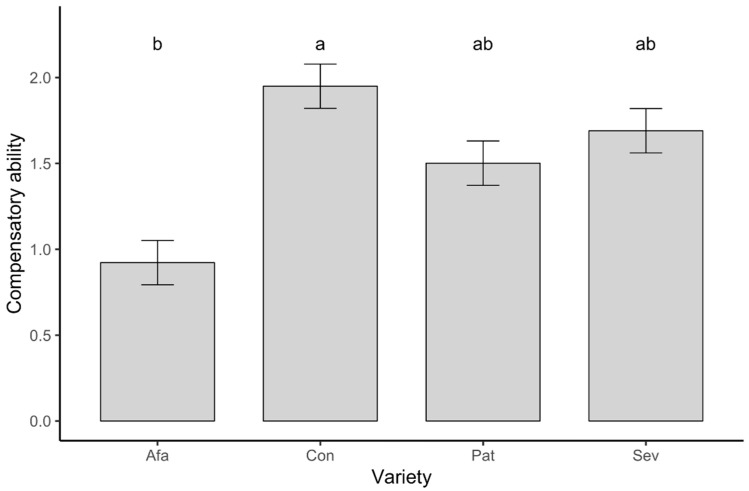
Least squares mean compensatory ability of plants of four tomato varieties. Letters indicate significant differences among means according to a Tukey’s multiple comparisons test.

**Figure 7 plants-09-01096-f007:**
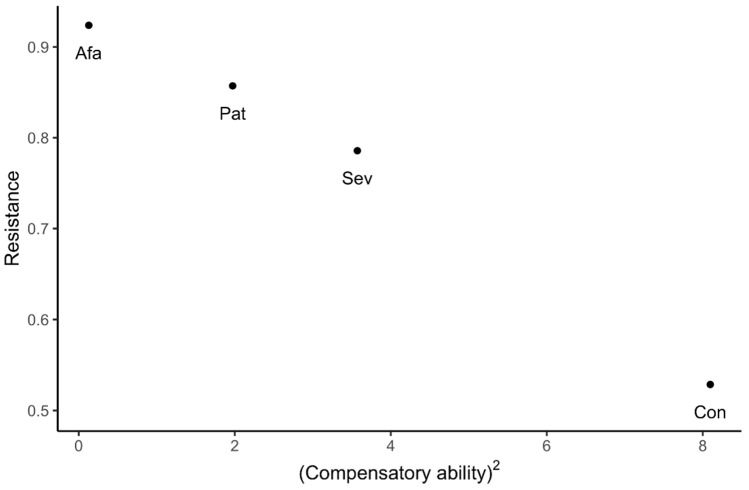
Mean resistance per variety as a function of the squared mean compensatory ability per variety.

**Table 1 plants-09-01096-t001:** Summary statistics (*F* and *p* values) of general linear model analysis for the effects of whitefly presence (W), variety (V), N level (N), and their interactions on total fruit fresh weight, total fruit dry weight, fruit size in fresh weight, seed production (full and reduced models), total seed mass, total vegetative biomass, and root:shoot ratio of plants in four tomato varieties grown at three N levels in greenhouse conditions.

Response Variable	W	C(W)	V	N	V*N	W*V	W*N	W*V*N
Total fruit fresh weight	0.75	3.33 *	0.76	20.17 ***	0.51	0.24	0.85	1.42
Total fruit dry weight	1.80	2.71 ^†^	2.29	19.17 ***	1.63	0.12	0.50	1.05
Fruit size (fresh weight)	0.15 ^a^	0.72	1.20	3.94 *	3.05 **	0.23	0.79	0.38
Seed production (full model)	1.70	2.66 ^†^	2.43 ^†^	2.95 ^†^	0.43	0.56	0.29	1.54
Seed production (reduced model)	1.76	2.58 ^†^	2.40 ^†^	3.21 *	- - -	- - -	- - -	- - -
Total seed mass	0.39	1.58	5.02 **	1.43	1.16	0.24	0.81	0.78
Total vegetative biomass	0.99	1.62	2.94 *	68.03 ***	1.31	1.16	1.21	0.64
Root: shoot ratio	1.07	3.06 *	27.35 ***	14.22 ***	0.83	0.42	0.45	0.77

*p* value: ^†^
*p* < 0.1; * *p* < 0.05; ** *p* < 0.01; *** *p* < 0.001. ^a^ not an exact *F* test.

**Table 2 plants-09-01096-t002:** Costs of chemical inputs and labor involved in applying those products for tomato production at an agronomically recommended rate (‘Normal’) and at half of that rate (‘Reduced’). All prices in US dollars (see text for details).

	Chile		USA	
*Cost of fertilizer*	**Normal**	**Reduced 50%**	**Normal**	**Reduced 50%**
Price of fertilizer per kg N	$1.06	$1.06	$0.62	$0.62
N application rate (kg N/ha) ^1;2;3^	520	260	252.2 ^4^	126.1
Expense on fertilizer per ha ^5;6^	$551.20	$275.60	$156.36 ^8^	$78.18
*Cost of pesticide per ha*	**Normal**	**Reduced 30%**	**Difference**	
Price of pesticide ^5; 6^	$1538.50	$1076.95		
Labor cost of application	$769.20	$538.44		
Total cost of pesticide application	$2307.70	$1615.39	$692.31	
*Cost of washing fruit*	**Normal**	**Reduced 15%**	**Difference**	
Tomato production kg/ha ^7^	63,000 ^7^	53,550		
Fruit that need to be washed (%)	33.0	28.05		
Cost of washing per kg	$0.034	$0.034		
Total cost of washing per ha	$706.86	$510.71	$196.15	
*Value of crop (gains)*	**Normal**	**Reduced 15%**	**Difference**	
Tomato production kg/ha	63,000 ^7^	53,550		
Value of crop/kg	$0.1253	$0.1253		
Gains per ha	$7894.21	$6710.08	$1184.13	
*Net gains or losses per ha*	**Gains**	**Losses**	**Net Change**	
Crop value		$1184.13		
Savings on N	$275.60			
Savings on pesticide	$692.31			
Savings on costs of washing fruits	$196.15			
Total	$1164.06	$1184.13	−$20.07	

^1–8^ References available in Appendix C (Appendix A).

**Table 3 plants-09-01096-t003:** Composition of modified Hoagland’s nutrient solution.

	Name	Molecular Weight	Concentration
**Macronutrients**			
K_2_HPO_4_	Potassium phosphate (monobasic)	136.6	1.8 mM
K_2_SO_4_	Potassium sulfate	174.26	3.6 mM
Ca(NO_3_)_2_·4H_2_O	Calcium nitrate (tetrahydrate)	236.4	4.45/2.2/0 mM *
MgSO_4_·7H_2_O	Magnesium sulfate (heptahydrate)	246.48	0.5 mM
CaCl_2_· 2H_2_O	Calcium chloride (dihydrate)	147.01	0/2.2/4.45 mM *
**Micronutrients**			
H_3_BO_3_	Boric acid	61.83	23 µM
MnCl_2_·4H_2_O	Manganese chloride	197.9	5 µM
ZnSO_4_·7H_2_O	Zinc sulphate	287.5	0.4 µM
CuSO_4_·5H_2_O	Cupric sulphate	249.7	0.2 µM
MoO_3_	Molybdic acid		0.1 µM
FeEDTA	Ethylene Diamine Tetraacetic Acid (ferric-sodium salt)	367.1	7 µM

* Respective concentrations used for the High, Medium, and Low N treatment levels.

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
