# Peer review of "Lower Nitrogen Availability Enhances Resistance to Whiteflies in Tomato"

_plants, 2020, doi:10.3390/plants9091096_

Round 1

Reviewer 1 Report

General comments:

The manuscript describes an interesting question of the difference between tolerance and resistance in tomato plants under different nitrogen levels. Overall it is a good manuscript and it clearly describes the purpose of the study.

My largest issue with the paper is the choice that the authors have taken about the presentation of the results. While it is generally a good idea to not overload a manuscript, there are a couple of additional figures that I would appreciate. Specifically

Figure 3: The corresponding figures for (a) and (b) with the varieties and N levels, respectively.

S2-S4, S7-S7: Show the statistical significance of effect of whiteflies (and interaction with other factors) on fruit weight etc. However none of these are mentioned in the manuscript. Wouldn’t it be important for a grower to know if fruit weight is effected differently by whitefly presences depending on N-values? You don’t necessarily need a figure for all of them, but maybe at least mention it in the manuscript specifically, instead of just having a disclaimer at the beginning of the result section that none of them were significant.

Specific comments:

l 142 (figure 3): y-axis of (a) add “number of”

l 142 (figure 3): as mentioned above add corresponding figures for (a) and (b)

l 143 (figure 3): you write “means ± 1 s.e.”. In figure 1 you wrote that error bars are standard error unless stated otherwise… is this standard error?

l 147: Why is the CI for vegetative biomass different? I could not find a description of it in the methods section.

l 183 (figure 4): add something along the lines of “higher values mean higher resistance”. Makes it more convenient for the reader.

l 260: should “generalists” and “specialists” be swapped?

l 272: spelling of “Moreover”

l 344: I suggest to add that “the amount of plants for each variety*N-level*whitefly combination was 10.” to make it more convenient for the reader.

l 348: remove the comma

l 380: I suggest to add that “the amount of plants of each variety*N-level combination per cage was 5.” to make it more convenient for the reader.

l 410 and l 413: How were they dried?

l 449: Where does the six come from?

l 558-460: Where are the results for these?

I hope these comments help to improve the manuscript.

Author Response

R1

Dear reviewer, thank you for your comments. They have helped us improved our paper. 

Please see our responses in the attached document

Reviewer 2 Report

The manuscript reports the outcomes of a short-term, small-scale, tri-factorial experiment addressing effects of nitrogen (3 levels) and whiteflies (2 levels) on biomass of tomato plants and plant resistance/tolerance to whiteflies. The experiment is well designed, and data are properly collected and analyzed. The authors also compared cost/benefit ratio between different strategies of fertilization and insecticide application. However, the manuscript is too long relative to the amount of data it contains, and suffers from some presentation problems.

Specific comments.

  1. Only one-third of the abstract describes the results of the study. I suggest focusing on the results, i.e. increasing their share in the abstract to two-thirds, accompanied by decrease of the total length of the abstract to 2/3 of its current volume.
  2. This nice case study does not require a comprehensive introductory review, and vote-counting is not the best way to review the relevant literature. I suggest removal of (i) appendix A listing 55 relevant studies and (ii) the respective parts of introduction. This will make a study more focused.
  3. The hypotheses concern whitefly resistance/tolerance only, whereas one of the obvious motivations for this study, namely cost/benefit ratio, is not mentioned among study goals.
  4. The root/shoot allocation hypothesis: if it refers to differences among varieties, then it is identical to the ‘d’ hypothesis. If, however, it refers to among-plant variation within varieties, then its importance remains obscure.
  5. The full results of statistical analyses should be included in the main document. I suggest to combine tables S2-S8 and report F/P values for each response variable as columns in this combined table.
  6. The results of statistical analyses given in the text do not support statements on differences among individual treatments, and only confirm the existence of among-treatment variation.
  7. Captions of Figs 1 and 2 do not explain error bars.
  8. Line 125: The non-significant results are not worse than the significant ones – the authors should not discriminate against these results and report (illustrate) them in the same way as they report (illustrate) significant results.
  9. A matter of curiosity: why the study plants produce to small fruits? Is this a norm for these varieties?
  10. Fig. 4: roots are also vegetative organs – please specify ‘vegetative biomass’ more precisely. Do ‘shoots’ include leaves and fruits?
  11. Discussion is too long and too general for this simple case study. I suggest making it more specific.
  12. The calculations of gains and losses form a nice part of the study. I suggest that this part is separated from the scientific discussion, e.g. by introduction of subheadings 3.1 Plant resistance and tolerance vs 3.2 Gains and losses associated with fertilizer and pesticide application.
  13. Lines 272-273: correct two typos.
  14. I doubt that Table 2 is really necessary. 

Author Response

R2

Dear reviewer, thank you for your comments. They have helped us improved our paper. Our responses are in the attached document.
